# Ectopic Expression of *BcCUC2* Involved in Sculpting the Leaf Margin Serration in *Arabidopsis thaliana*

**DOI:** 10.3390/genes14061272

**Published:** 2023-06-15

**Authors:** Wanqi Li, Tongtong Wang, Yu Ma, Nan Wang, Wenjing Wang, Jun Tang, Changwei Zhang, Xilin Hou, Hualan Hou

**Affiliations:** 1School of Horticulture, Anhui Agricultural University, Hefei 230036, China; 2College of Horticulture, Nanjing Agricultural University, Nanjing 210095, China; 3Jiangsu Key Laboratory for Horticultural Crop Genetic Improvement, Jiangsu Academy of Agricultural Sciences, Nanjing 210014, China

**Keywords:** Pak-choi, *BcCUC2*, leaf serration, expression patterns, *Arabidopsis thaliana*

## Abstract

Leaf margin serration is a morphological characteristic in plants. The *CUC2* (*CUP-SHAPED COTYLEDON 2*) gene plays an important role in the outgrowth of leaf teeth and enhances leaf serration via suppression of growth in the sinus. In this study, we isolated the *BcCUC2* gene from Pak-choi (B*rassica rapa* ssp. *chinensis*), which contains a 1104 bp coding sequence, encoding 367 amino acid residues. Multiple sequence alignment exhibited that the *BcCUC2* gene has a typical conserved NAC domain, and phylogenetic relationship analysis showed that the *BcCUC2* protein has high identity with *Cruciferae* plants (*Brassica oleracea*, *Arabidopsis thaliana*, and *Cardamine hirsuta*). The tissue-specific expression analysis displayed that the *BcCUC2* gene has relatively high transcript abundance in floral organs. Meanwhile, the expression profile of *BcCUC2* was relatively higher in the ‘082’ lines with serrate leaf margins than the ‘001’ lines with smooth leaf margins in young leaves, roots, and hypocotyls. In addition, the transcript level of *BcCUC2* was up-regulated by IAA and GA3 treatment, especially at 1–3 h. The subcellular localization assay demonstrated that BcCUC2 was a nuclear-target protein. Furthermore, leaf serration occurred, and the number of the inflorescence stem was increased in the transgenic *Arabidopsis thaliana* plants’ overexpressed *BcCUC2* gene. These data illustrated that *BcCUC2* is involved in the development of leaf margin serration, lateral branches, and floral organs, contributing to further uncovering and perfecting the regulation mechanism of leaf serration in Pak-choi.

## 1. Introduction

Leaves are important vegetative organs in plants, and their shape and size directly affect the photosynthesis, transpiration, stress response, and ornamental value of plants. Leaf morphogenesis starts from the flanks of a small group of totipotent stem cells, the shoot apical meristem (SAM) [1]. The development of leaves can be divided into three main stages: (1) the initiation of leaf primordium; (2) the establishment of primary leaf shape, the leaf primordium continuing to grow and differentiate, and the production of secondary structures such as serrate leaves, lobed leaves, and leaflets; (3) the formation of secondary leaf morphology, producing leaf margins, stomata, trichome, and eventually forming mature leaves [2,3].

Plant leaves can be defined as entire leaves (smooth margins), serrate leaves, and lobed leaves according to the margins of the leaf and leaflet blades [4]. Leaf morphology is complex and varied in diverse species, which mainly depends on the regulation of genetic, developmental, and environmental factors [5]. So far, a number of leaf margin regulators have been identified with crucial roles in elaborating leaf shape. The *KNAT1*, *KNAT2*, and *SHOOT MERISTEMLESS* (*STM*) genes belong to *KNOTTED*-*like homeodomain* class I (*KNOX1*) transcription factors, which are involved in the formation and maintenance of the SAM. The expression level of *KNOX1* genes is down-regulated during early leaf initiation, and the overexpression of *KNAT1* and *KNAT2* leads to occasional ectopic shoots on the adaxial surface of leaves, lobed leaves, and ectopic stipules [6,7,8,9,10]. In *Arabidopsis thaliana* (*A. thaliana*) (L.) Heynh., the loss of function of *STM* is associated with the evolution of the unlobed leaf form; thereby, *STM* is essential for lobe formation [11].

*CUP*-*SHAPED COTYLEDON* (*CUC*), members of the NAC transcription factors (such as NAM in *Petunia hybrida* and *ATAF1*/*2* and *CUC2* in *A. thaliana*), which contain a conserved NAC domain at the N-terminus and a highly variable domain in the C-terminal region [12], act as major players in shoot apical meristem (SAM) construction, organ separation, leaf development, and the regulation of the axillary meristem initiation in leaf axils [4,13,14,15,16,17]. In *Arabidopsis*, the CUC subfamily contains three members, *CUC1*, *CUC2*, and *CUC3*, which act redundantly to regulate cotyledons’ separation, organ boundary specification, and embryonic shoot meristem formation, in part [13,16,18]. The *CUC1* and *CUC2* genes are necessary for shoot meristem initiation via promoting the transcript level of *STM* [19]. The *cuc1* and *cuc2* single mutants display few morphological phenotypes due to their functional redundancy, while the *cuc1cuc2* mutant exhibits complete absence of shoot meristem and forms dramatically fused cotyledons [13,18,20]. Mutation of the *CUC* homologs, the *CUPULIFORMIS*, *NO APICAL MERISTEM* (*NAM*), and *GOBLET* genes in snapdragon, petunia, and tomato, respectively, results in similar development defects [21,22,23,24], illustrating that these genes share an evolutionarily conserved function in organ separation and SAM development. *CUC2* plays an important role in the initiation of leaf serration during the early phase and has a synergistic interaction with *CUC3* in the maintenance of leaf serration during the later stage [15]. In addition, *MIR164A* encodes a microRNA, which is involved in the regulation of leaf margin serration through cooperating with *CUC2* in *Arabidopsis*. The *mir164a* mutants display significant deep serrate leaf margins compared with the wild type in *Arabidopsis* [4]. The *CUC2* gene is one of the target genes of *miR164*, and the balance between *CUC2* and *miR164a* has a prominent role in the extent of leaf serrations. In *Solanum lycopersicum*, loss of *GOBLET* (*GOB*), the homologous gene of *Arabidopsis CUC2*, results in reduced complexity of compound leaves and fruit shape [24], suggesting that the *CUC2* gene has diverse functions in different species and plays a crucial role in leaf development.

Pak-choi (*Brassica rapa* ssp. *chinensis*) belongs to *Brassicaceae* crops, is an important leafy vegetable, and is widely cultivated in the middle and lower regions of the Yangtze River. Leaf morphology, as an important agronomic trait, has a direct effect on its yield and ornamental value. In this study, we isolated a *CUC2* gene encoding 367 amino acids from Pak-choi, which has three highly conserved *CUC2*-specific motifs. The expression patterns and biological function of *BcCUC2* were systematically investigated.

## 2. Materials and Methods

### 2.1. Plant Materials and Growth Conditions

Pak-choi cultivar ‘001’ with entire leaf margin and ‘082’ with serrate leaf margin were used in this study. Seeds were dispersed on wet filter paper for germination and grown in pots containing humus soil/vermiculite (2:1) mixture in plant artificial climate chamber controlled at 23/17 °C, 16/8 h for light/dark cycle. The illumination intensity was set to 12,000 xl, and the relative humidity was 65–75%. For different tissues, the samples were harvested at seedling, rosette, flowering, and podding stages. Three biological replicates were used for each sample.

*A. thaliana* wild type (*Columbia*-0) was used in this study and grown in illumination incubator under the same condition. Four-week-old tobacco (*Nicotiana benthamiana*) seedlings were used for subcellular localization assay.

### 2.2. IAA and GA3 Treatments

Seedlings of cultivar ‘001’ with six fully expanded leaves were foliar sprayed with 100 μM GA3 and IAA. The leaves were collected at 1 h, 3 h, 6 h, 12 h, 24 h, and 48 h after treatment (0 h was used as a control) and frozen rapidly in liquid nitrogen, followed by storage at −70 °C in refrigerators for RNA extraction. Three biological repeats were implemented for each sample.

### 2.3. Cloning and Sequence Analysis

Total RNA was extracted from leaves using RNAprep pure Plant Kit (TIANGEN, Beijing, China), and the first strands of cDNA were synthesized via reverse transcription using PrimeScript™ II 1st Strand cDNA Synthesis Kit (Takara, Dalian, China). The coding sequence (CDS) of *BcCUC2* was amplified by gene-specific primers (see Table A1) based on the sequence of *Bra022685* using homology cloning referred to in our previous report [25]. The resulting fragment was cloned into pEASYBlunt Simple Vector (Transgene, Beijing, China) and sequenced by Genscript Company (Nanjing, China). The physicochemical characteristics of *BcCUC2* were predicted using Expasy proteomics server (https://web.expasy.org/protparam/, accessed on 10 July 2020). The secondary structure of *BcCUC2* was predicted using PSIPRED 4.0 (http://bioinf.cs.ucl.ac.uk/psipred/, accessed on 10 July 2020). 

### 2.4. Phylogenetic Tree Analysis

The protein sequences of *A. thaliana* were obtained from TAIR database (http://arabidopsis.org/index.jsp, accessed on 15 July 2020). The protein sequences of *Brassica oleracea*, *Brassica hirsuta*, *Solanum lycopersicum*, *Oryza sativa*, *Zea mays*, and *Vitis vinifera* were downloaded from NCBI database (https://www.ncbi.nlm.nih.gov/, accessed on 15 July 2020). Multiple sequence alignment of *CUC* genes in different species was performed using ClustalW software. The phylogenetic tree was constructed by MEGA 7.0 using neighbor-joining (NJ) method with the bootstrap of 1000 replications. All protein sequences used in this study are listed in Table A2. The conserved motifs distribution was analyzed by the MEME program, and the number of motifs was set as 10 (Figure 1B).

### 2.5. Subcellular Localization Assay

The full-length CDS of *BcCUC2* without stop codon was subcloned into linear pRI101-GFP vector with *NdeI* and *BamHI* restriction enzyme to generate the construct 35S:*BcCUC2*-GFP. For transient expression assay, empty vector (35S:GFP) and recombinant construct were introduced into *Agrobacterium tumefaciens* GV3101 and injected into tobacco foliar epidermis, respectively. After dark culturing for 24 h, the seedlings were moved to normal growth conditions for 24–36 h, and then fluorescent images were photographed using a confocal laser scanning microscope (Zeiss, LSM780, Jena, Germany). 

### 2.6. Ectopic Expression in Arabidopsis

The coding sequence of *BcCUC2* was inserted into the pCAMBIA1301 vector to produce the construct 35S:*BcCUC2*-GUS; then, the recombinant construct was transformed into *Arabidopsis* using the floral dipping method via *Agrobacterium*-mediated transformation [26]. In brief, the 35S:*BcCUC2*-GUS plasmids were transferred into *A. tumefaciens* GV3101 and agroinfiltrated into the *Arabidopsis* flowers for 45–60 s, cultivated in dark conditions for 2 d, and then grown in plant artificial climate chamber. To obtain the positive overexpression lines, seeds of *Arabidopsis* transgenic plants were surface sterilized and sown on half-strength Murashige and Skoog (1/2 MS) tissue culture plates containing 30 mg/L of hygromycin. The resistant seedlings were further testified using gene-specific primers amplification and GUS staining. Finally, 20 plants for each T3 positive transgenic line were cultivated and used for phenotype analysis.

### 2.7. Quantitative Real-Time PCR

Total RNAs were extracted using RNAprep pure Plant Kit (TIANGEN, Beijing, China) according to the operation manual, and cDNA was synthetized with PrimeScript™RT reagent Kit with gDNA Eraser (Perfect Real Time) (Takara, Dalian, China). qRT-PCR was performed with ABI StepOnePlus™ Real-Time PCR System (Applied Biosystems, Foster City, CA, USA) using TransStart Tip Green qPCR SuperMix (TransGen, Beijing, China). The PCR procedure was carried out with the following parameters: 94 °C for 30 s and 40 cycles of 94 °C for 5 s, 60 °C for 15 s, 72 °C for 10 s. The *Actin* genes of Pak-choi and *Arabidopsis* were used as internal control, and the relative expression levels were calculated utilizing the 2^−ΔΔCt^ method [27]. All primers used in this study are listed in Table A1.

## 3. Results

### 3.1. Cloning and Characteristic Analysis of BcCUC2

The full-length fragment of the *BcCUC2* gene was amplified from Pak-choi cultivar ‘001’ using primer pair *BcCUC2*-F and *BcCUC2*-R. The BcCUC2 protein is an unstable and hydrophilic protein (GRAVY = −0.567), encoding 367 amino acid residues; the theoretical isoelectric point (pI) is 8.47, and the molecular weight is 40.97 KDa via Expasy online prediction. The BcCUC2 protein has a typical conserved NAC domain at the N-terminal (18–144 amino acid sites) through conserved domain search analysis in NCBI, and it belongs to the NAC family of plant-specific transcription factors. Meanwhile, multiple sequence alignment analysis revealed that the CUC protein sequence of different species shares a highly conserved NAC domain at the N-terminal, while the C-terminal is a transcriptional activation region with abundant variation, reflecting the diversity of species, which is also a general structural feature of NAC transcription factors (Figure 1A). Furthermore, multiple sequence alignment implied that the BcCUC2 protein shares 38.61% and 48.5% identity with *OsNAM* from rice and *SlNAM* from tomato, while sharing 81.38% and 74.73% identity with *AtCUC2* from *Arabidopsis* and *ChCUC2* from *Cardamine hirsuta*, suggesting the conservation of *Cruciferae* species in evolution. Meanwhile, the MEME analysis indicated that motifs 1/2/3/6 were extremely conserved sequences in all species, while motifs 7/9/10 were CUC2-specific sequences of *Cruciferae* plants (Figure 1B). Additionally, the secondary structure analysis showed that the BcCUC2 protein was mainly composed of α helices, random coils, and extended strands (Figure A1).

### 3.2. Phylogenetic Tree Analysis of BcCUC2

To explore the phylogenetic relationship of *CUC* genes, an unrooted phylogenetic tree was constructed using MEGA 7.0 software with the neighbor-joining (NJ) method. As shown in the phylogenetic tree (Figure 2), the *CUC* genes were firstly divided into two clades, the *NAM*/*CUC1*/*CUC2* clade and the *CUC3* clade. Meanwhile, the *CUC* genes from dicots (*A. thaliana*, *C. hirsuta*, *S. lycopersicum*, *V. vinifera*, *B. oleracea*, and Pak-choi) were clustered together and separated from monocots (*O. sativa* and *Z. mays*) into two clades, and the dicot–monocot split occurred 150 million years ago (Mya) [28]. Furthermore, *BcCUC2* has a higher similarity with the *B. oleracea CUC2* gene (*Brassica*), followed by the *A. thaliana* and *C. hirsuta CUC2* gene (*Cruciferae*), and *Brassica rapa* diverged from *A. thaliana* and *B. oleracea* at 20 Mya and 8 Mya, respectively [29,30]. The result indicated that the molecular evolutionary relationship of the CUC2 protein between Pak-choi and seven other species is basically consistent with the genetic relationship.

### 3.3. Expression Pattern Analysis of BcCUC2 in Pak-choi

The NAM/CUC3 subfamily plays an important role in the formation of shoot meristem and boundary, leaf margin serration, compound leaf, and axillary meristem (lateral branch) [4,13,14,15]. To mirror the spatiotemporal expression patterns of *BcCUC2* in Pak-choi, quantitative RT-PCR analysis was carried out. As shown in Figure 3, *BcCUC2* was expressed relatively higher in petioles at the seedling stage, in stems and roots at the rosette stage, and in floral organs at the flowering and podding stages, while being weakly expressed in leaves and hypocotyls in cultivar ‘001’ with entire leaf margins.

In addition, to investigate the function of *BcCUC2* in leaf morphology, we comprehensively examined the transcript levels of the *BcCUC2* gene in cultivar ‘082’ with a serrate leaf margin. The result indicated that *BcCUC2* has higher expression in leaves, roots, and hypocotyls at the seedling stage, stems at the rosette stage, and pods and floral organs at the flowering and podding stages. The transcript levels of *BcCUC2* were significantly higher in the young leaves of cultivar ‘082’ than in cultivar ‘001’, while being decreased in mature leaves, suggesting a possibility that *BcCUC2* has a role in the early steps of leaf serration. Therefore, we inferred that *BcCUC2* may participate in leaf morphogenesis and flower development.

### 3.4. Expression Analysis of BcCUC2 Gene under Hormone Treatment

Several studies have shown that auxin and gibberellin have an important role in the regulation of leaf shape development [31,32,33]. To dissect the response of *BcCUC2* to auxin and gibberellin, quantitative expression analysis was performed for the *BcCUC2* gene under IAA and GA3 treatment. As shown in Figure 4, the *BcCUC2* gene was highly expressed under exogenous IAA and GA3 treatment at the initial stage and peaked at the 3 h time point, demonstrating that the *BcCUC2* gene may be regulated by IAA and GA3 through the means of promotion.

### 3.5. Subcellular Localization Analysis of BcCUC2

To confirm whether the *BcCUC2* gene, as a putative transcription factor, is localized in the nucleus, the 35S:*BcCUC2*-GFP fusion protein was constructed and used for transient transformation in *Nicotiana benthamiana* using *Agrobacterium tumefaciens*-mediated transfection methodology. The laser confocal scanning microscope images of tobacco epidermal cells showed that the green fluorescence of the 35S:*BcCUC2*-GFP fusion protein was mainly distributed in the cell nucleus, while the empty vector (35S:GFP protein, negative control) was expressed in the whole cell, illustrating that BcCUC2 is a nuclear-localized protein (Figure 5).

### 3.6. Ectopic Expression of BcCUC2 Caused Leaf Margin Serration and Increased Lateral Branches in Arabidopsis

In order to explore the potential function of *BcCUC2*, we first transformed the *BcCUC2* gene into *Arabidopsis*. The positive transgenic plants were authenticated by PCR amplification using specific primers and GUS staining, and the qRT-PCR was used to check the gene expression level for further verification (Figure A2). Finally, thirteen transgenic lines (termed as OE1-OE13) were obtained, and four lines (OE5, OE6, OE9, OE11) with relatively higher expression levels of *BcCUC2* were selected for further research (Figure 6B). We observed that leaf margins were modified when the *BcCUC2* gene was overexpressed in *Arabidopsis*. The phenotype of leaf margin serration was significantly presented in transgenic *Arabidopsis* plants expressing the *BcCUC2* gene, which were clearly distinct from the leaves of wild-type *Arabidopsis* (Figure 6C). In addition, the ectopic expression lines with the *BcCUC2* gene displayed a significant increase in the number of inflorescence stems, flowers, and siliques in comparison with the wild type (Figure 6D). These results demonstrated that *BcCUC2* plays critical roles in the formation of leaf margin serration and the development of flower and lateral branches, which is consistent with the high expression abundance in stem and floral organs.

## 4. Discussion

Previous studies have shown that the serrate morphology of deep-lobed leaf blades is conducive to heat dissipation and defense against high-temperature burns, thereby improving the survival probability of plants [34]. The hydraulic efficiency of the deep-lobed leaves is high, which enhances their adaptability in arid environments [35]. Meanwhile, deep-lobed morphology makes the leaves have a larger specific leaf area in space and thus has stronger competition for light resources and higher photosynthesis efficiency than entire leaves [36]. In addition to its adaptive ability in diverse conditions, the degree of dissection of the leaf margins may add to the ornamental value of leaves in different species. The functional analysis of genes associated with leaf serration development contributes to improving the leaf margin traits using biotechnological methods. Leaf shape is a major agronomic trait, and leaf margins can be lobed, serrate, or entire in plants. Pak-choi, as an economically leafy vegetable, has important edible value and is widely cultivated in Asia. However, the candidate genes and molecular mechanism of serrate leaf margins remain to be fully elucidated.

The NAM/CUC subfamily, including *CUC1*, *CUC2*, and *CUC3*, belongs to the NAC (NAM, ATAF1/2, and CUC) transcription factors family, playing a central and redundant role in plant organ development and organ boundary formation, e.g., in floral organs (gynoecium and ovules), leaf serration, and primary and axillary shoots (reviewed in [37]). So far, *CUC* genes have been widely studied in several species. For example, in *Arabidopsis*, *CUC1* and *CUC2* are involved in the formation of carpel margin meristem by controlling shoot meristem activity [38] and regulating carpel margin development through interacting with *SPATULA* (*SPT*) [39], and *CUC3* has significant roles in regulating organ boundary formation and postembryonic shoot meristem [16]. In addition, *CUC2* participates in ovule primordia formation via direct interaction with the DELLA protein GAI [33]. The balance between *miR164a* and *CUC2* is responsible for the extent of leaf serrations [4]. In strawberry, the *miR164*-*CUC2* regulatory module plays conserved and novel roles in specifying leaf and floral organ morphology [40]. In tomato, the *GOBLET* (*GOB*) gene, homologous to the *CUC2* gene, plays important roles in regulating fruit shape and the complexity of compound leaves [24]. In *Liriodendron chinense*, the *LcCUC2*-*like* (*LcCUC2L*) gene, homologous to *AtCUC2* in sequence, has an important role in controlling cotyledon development and rosette leaf number [32]. According to the above reports, the *CUC2* gene has prominent roles in organ boundary formation and organ number.

The functional importance and relationship of the *CUC2* gene in Pak-choi, however, is poorly understood. Here, we performed systematic analysis of the characterization and function of *BcCUC2* using bioinformatics tools, real-time PCR, and ectopic expression in *Arabidopsis*. The phylogenetic tree displayed that the *BcCUC2* gene was clustered into the NAM/CUC1/CUC2 clade and close to the *B. oleracea BoCUC2* and *A. thaliana AtCUC2*, which all belong to *Cruciferae*, consistent with their evolutionary history. The tissue-specific expression analysis showed that *BcCUC2* has relatively higher transcript levels in stem and floral organs and differential expression in leaves with smooth margins or serrate margins, indicating that *BcCUC2* may be involved in the proper control of leaf morphology and the development of floral organs. In addition, the expression abundance of the *BcCUC2* gene was observably greater in young leaves with serrate margins (cultivar ‘082’) than smooth margins (cultivar ‘001’), while being markedly decreased in mature leaves, possibly implying that *BcCUC2* participated in the formation of serrate margins in early leaf development. In the *Pro35*:*BcCUC2* lines, the transcript levels of *BcCUC2* were significantly up-regulated and the phenotype of leaf serration and increased inflorescence stems, flowers, and siliques were found, supporting our hypothesis that *BcCUC2* is involved in modulating the development of floral organs and leaf morphology. It is interesting that the phenotype of leaf serration was significant at the seedling stage, while being diminished at mature and flower stages in transgenic lines, which is consistent with the spatiotemporal expression profile of *BcCUC2*. On the other hand, given that the overexpression of *BcCUC2* in *Arabidopsis* resulted in non-significant enhancement of serration, it is possible that *BcCUC2* may be less functional in *Arabidopsis* than in Pak-choi, or line ‘082’ may have another potential gene to enhance serration.

In plants, phytohormones have a crucial role in the growth and development and cell morphogenesis of diverse tissues. Leaf shape traits are regulated by an intricate regulatory network concerning transcription factors and hormone signaling [4]. For instance, *PIN1*, as an auxin efflux carrier, plays an essential role in auxin distribution in the placenta, and the expression of *PIN1* was up-regulated by *CUC* genes, which have crucial roles in leaf shape [31]. In *L. chinense*, *LcCUC2L* regulates leaf development by regulating the auxin content [32]. Furthermore, the ovule primordia formation was modulated by *CUC2’s* direct interaction with the gibberellin signaling protein GAI [33]. To investigate the response of *BcCUC2* to auxin and gibberellin, the expression patterns of *BcCUC2* under IAA and GA3 treatment were analyzed. The result showed that the transcript levels of *BcCUC2* were significantly increased at 1–3 h after treatment, indicating that *BcCUC2* responds to auxin and gibberellin. Based on the above results, we speculated that *BcCUC2* is involved in the formation of serrate leaf margins through the mediation of auxin or gibberellin. 

Although the expression patterns and potential function of *BcCUC2* have been preliminarily determined based on real-time PCR and overexpression assay, the specific reason for the difference in leaf morphology and *BcCUC2* expression profiles between the lines ‘001’ and ‘082’ remains unclear. Several studies have shown that *miR164* level or auxin levels have an essential role in the development of leaf morphology [4,31,32,40]. Thus, the promoter sequence of *BcCUC2*, *miR164* level, or auxin levels between the lines ‘001’ and ‘082’ will be analyzed to further elucidate the molecular mechanism of leaf morphology and *BcCUC2* expression difference between the lines.

## 5. Conclusions

In conclusion, we have isolated a *CUP*-*SHAPED COTYLEDON* (*CUC*) gene in Pak-choi named *BcCUC2*. The phylogenetic reconstruction displayed that the *BcCUC2* gene is clustered into the *CUC1*/*CUC2* clade and is close to the *B. oleracea BoCUC2* and *A. thaliana AtCUC2*, which all belong to *Cruciferae.* The expression patterns of the *BcCUC2* gene were significantly different in leaves and petioles between cultivar ‘001’ with entire leaf margins and ‘082’ with serrate leaf margins, while similar in floral organs. In addition, the transcript abundance of *BcCUC2* was significantly induced by IAA and GA3 treatment. The heterologous expression of *BcCUC2* resulted in leaves with serrate margins and more inflorescence stems.

## Figures and Tables

**Figure 1 genes-14-01272-f001:**
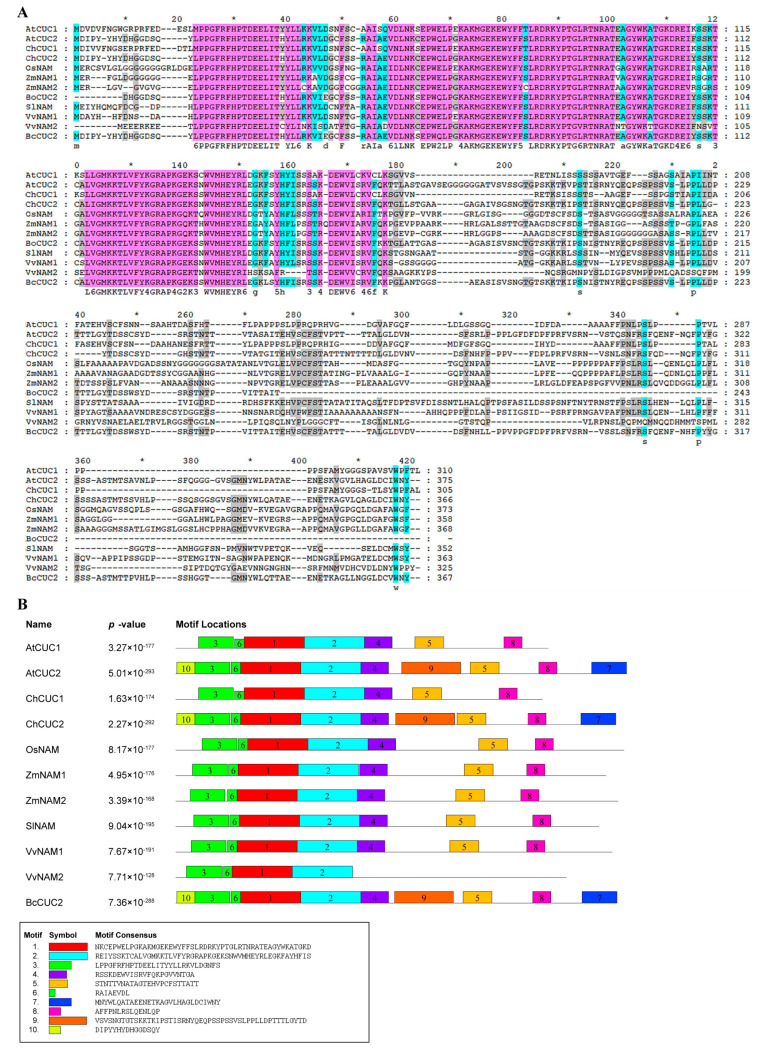
Alignment and conserved motif of plant CUC protein sequences. (**A**) Multiple sequence alignment of BcCUC2 with other CUC proteins. (**B**) Conserved motif analysis of CUC proteins.

**Figure 2 genes-14-01272-f002:**
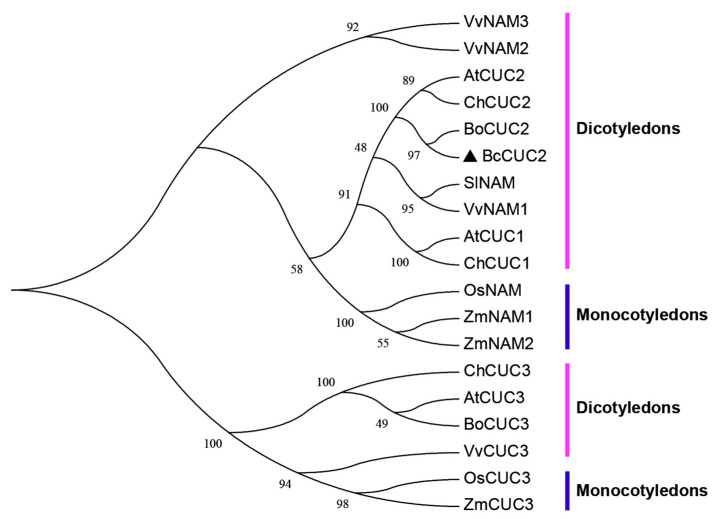
Phylogenetic analysis of *CUC* genes. Phylogenetic relationships between BcCUC2 and other CUC proteins in different species. The unrooted phylogeny was constructed by the neighbor-joining (NJ) method using MEGA 7.0 software.

**Figure 3 genes-14-01272-f003:**
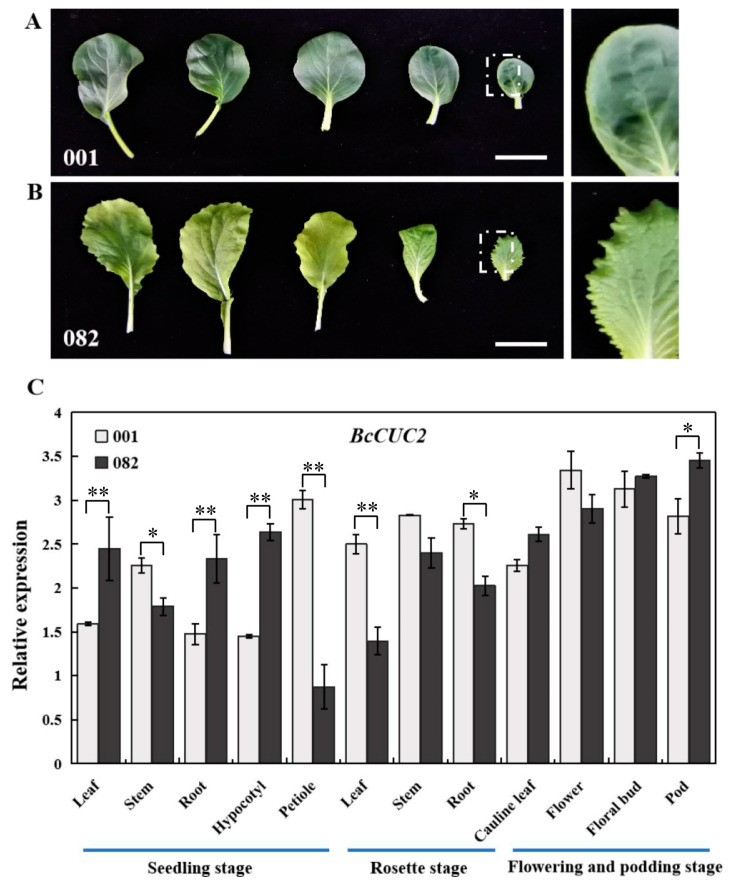
The transcript levels of *BcCUC2* in Pak-choi. (**A**,**B**) The leaf morphology of cultivars ‘001’ and ‘082’ at seedling stage. Scale bars: 5 cm. (**C**) The expression profile of *BcCUC2* at seedling, rosette, flowering, and podding stages in cultivars ‘001’ and ‘082’. The data represent the means of three replicates. A *t*-test was used for statistical analysis: * *p* < 0.05; ** *p* < 0.01.

**Figure 4 genes-14-01272-f004:**
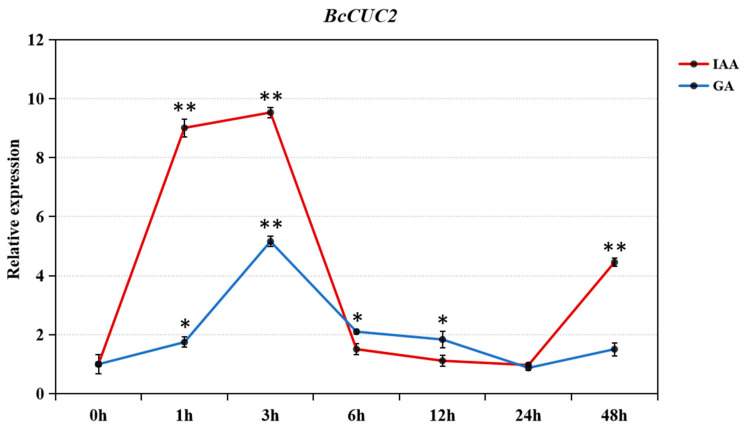
Relative expression patterns of *BcCUC2* gene under different hormone treatments. Six-leaf-stage Pak-choi plants (cultivar ‘001’) were subjected to IAA and GA3 treatments over a continuous time course (0, 1, 3, 6, 12, 24, 48 h). The transcript abundance of *BcCUC2* at 0 h was used as a control. Statistical significance (ANOVA) is designated by * *p* < 0.05, ** *p* < 0.01.

**Figure 5 genes-14-01272-f005:**
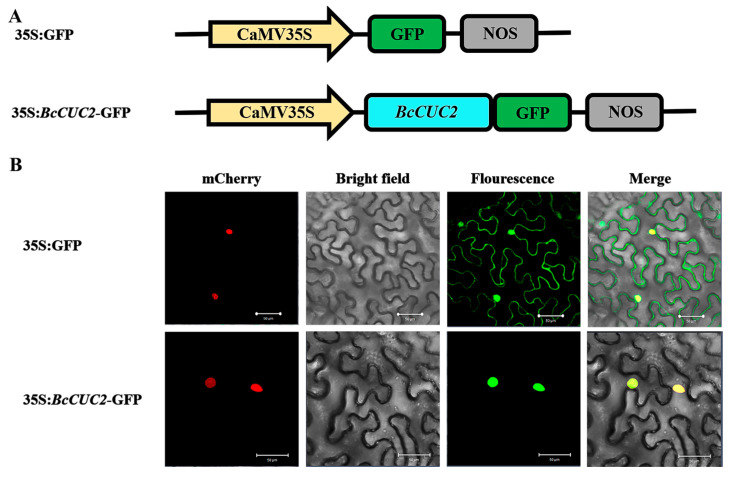
Subcellular localization of BcCUC2 protein. (**A**) The fusion constructs of 35S:GFP and 35S:*BcCUC2*-GFP. (**B**) Localization of 35S:GFP and 35S:*BcCUC2*-GFP in tobacco epidermic cells. The red, green, and yellow in the panel represent the fluorescence of mcherry (nuclear marker), GFP, and merge, respectively. Scale bars = 50 μm.

**Figure 6 genes-14-01272-f006:**
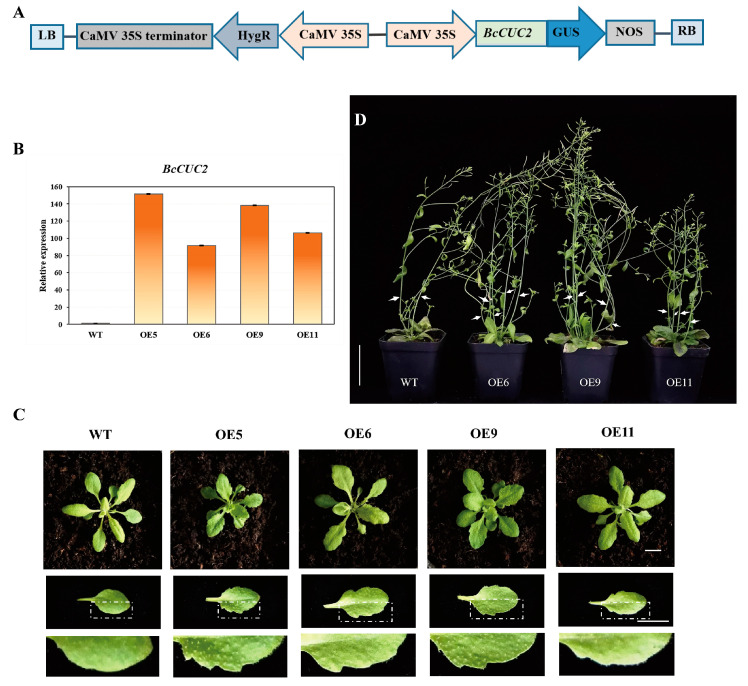
Ectopic expression of *BcCUC2* resulted in serrated leaves and many branches. (**A**) The construct of 35S:*BcCUC2*-GUS. (**B**) The relative expression level of *BcCUC2* in transgenic plants. (**C**) The leaf morphological difference in WT and transgenic plants. Scale bars = 1 cm. (**D**) Transgenic plants have more stem branches at flowering stage compared with wild type. Scale bar = 5 cm.

## Data Availability

The data supporting the findings of this study are available from the corresponding author upon reasonable request.

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
