# Peer review of "Ectopic Expression of BcCUC2 Involved in Sculpting the Leaf Margin Serration in Arabidopsis thaliana"

_genes, 2023, doi:10.3390/genes14061272_

Round 1
Reviewer 1 Report
In the reviewed manuscript entitled "Ectopic expression of BcCUC2 involved in sculpting the leaf margin serration in Arabidopsis thaliana" BcCUC2 gene from Pak-choi (Brassica rapa ssp. chinensis), was isolated and analysed using bioinformatics tools, real-time PCR, and ectopic expression in Arabidopsis. Pak-choi (Brassica rapa ssp. chinensis), belongs to Brassicaceae crops and is an important leafy vegetable and widely cultivated. BcCUC2 gene has a typical conserved NAC domain, and phylogenetic relationship analysis showed that BcCUC2 protein is highly similar to other Cruciferae species such as Brassica oleracea, Arabidopsis thaliana and Cardamine hirsuta. The tissue-specific expression analysis displayed this gene transcript abundance in floral organs. Expression profile analysis revealed that BcCUC2 expression was higher in the ‘082’ lines with serrate leaf margins than ‘001’ lines with smooth leaf margins. Moreover, the expression level of BcCUC2 was up-regulated by IAA and GA3 treatment. The data obtained by the Authors proved that BcCUC2 is involved in the development of leaf margin serration, lateral branches, and floral organs.
The manuscript was prepared very carefully. The introduction is thorough enough and contains the necessary information. The authors can only check whether the range of cultivation of this species is not limited to one region in China. The Materials and Methods chapter is described in detail enough. Although, of course, more details could be added, especially regarding the transformation. The results are also presented in a clear way, presented in figures and graphs. The discussion and conclusions were also properly conducted.
Additional comments are included in the attached file.

Minor editing of English language required.
Some grammar errors were pointed out in the attached file.
Reviewer 2 Report
CUC2 gene in many land plant species is known as a regulator for multiple traits related to meristem function, leaf morphology and reproduction, thus expected to be a potential genetic resource for yield improvement in crop.
This manuscript shows data suggesting BcCUC2 functionality in Pak-choi from view points of sequence homology, protein localization, hormone responsive expression and the effect of the overexpression in Arabidopsis. Their approaches are rational to get insights on Pak-choi leaf development and BcCUC2 function. They also show the different expression profile between two lines (001 and 082) with different leaf margin morphology suggesting possible natural variation in CUC2 regulation in Brassica rapa plants. In addition, negative regulation for CUC2 by auxin is known in Arabidopsis, but the data here show positive regulation in Pak-choi, which can be interesting from developmental and evolutionary points of view.
Taken together, I think the topic and some of the data may attract interests of readers from domains of plant development and crop breeding. Nevertheless, I think the manuscript does not meet the level of data quantity and quality for publication. The major points of my concern are listed below.
1. Little insight into regulatory mechanism of BcCUC2
Homology of the coding and protein sequences of CUC2 genes suggests that BcCUC2 is a functional ortholog. Transgenic overexpression in Arabidopsis showed developmental functions of BcCUC2. However, what makes difference of BcCUC2 expression profiles and leaf morphology between the lines 001 and 082?
For example, differences in promoter sequence of BcCUC2, miR164 level or auxin levels can result in the expression differences between the lines. Or it would be informative to compare BcCUC2 expression between line 001 and 082 after auxin and GA treatments.
Another approach would be transgenic experiments introducing genomic fragments containing BcCUC2 from line 001 and 082 to check the functional difference between the lines.
In addition, comparing BcCUC2 and AtCUC2 (and other CUC2 genes) in terms of promoter sequences or effects of overexpression may provide insights into CUC2 evolution.
I understand that it is not the focus of this paper to perform all of these experiments I am suggesting and to elucidate the regulatory variation of CUC2, but even a small amount of data would provide novel insights into the importance of BcCUC2 as a genetic resource and the gene evolution, which I think is the primary focus of this study.
2. The link is unclear between BcCUC2 expression profiles and the leaf morphology
Authors point out that differences in BcCUC2 expression profiles may explain the differences in leaf serration. However, it is also known that leaf morphology become smoother in the auxin transport mutant pin1 and the auxin response mutant iaa8 iaa9 in Arabidopsis, and auxin changes CUC2 expression levels in Arabidopsis and also in Pak-choi as the authors themselves show. From these facts, it is possible that the distribution and responsiveness of auxin is different between line 001 and 082, thereby causing differences in leaf margin morphology, and at the same time, the expression level of CUC2 is affected as a side effect. In fact, the difference in BcCUC2 expression levels between line 001 and 082 is less than 2-fold, with no p-values of significance. On the other hand, given that serrations of 35S:BcCUC2-GUS lines are only slightly developed after dozens of folds of overexpression, it is not very logical to discuss that differences in BcCUC2 expression profiles can be the major cause of the leaf serration differences.
3. Data sizes are not clearly described
In Figures 3, 4, and 5 BcCUC2 expression is quantified by qPCR, but the biological replicates are not specified. Fig. 5 also lacks information on the number of samples tested. Due to the lack of information on the data size, the reproducibility of the data cannot be adequately evaluated.
4. Lack of quantification of morphological traits
Although a small number of representative photos are shown in Fig. 1 and 6, it is not clear whether there are variation in the same genotype and differences from line to line, so it is not possible to determine whether they are statistically significant. There are multiple methods for quantifying the serration level of leaves, including simply counting the number of outgrowth, measuring the height of each individual serration, and quantifying the complexity of the contour (bending energy, solidity, etc.), which have been used in previous studies.
The authors also point out that there are more branches in the 35S:BcCUC2-GUS lines than WT, which they regard as providing an important insight into BcCUC2 function, but they do not count the number of branches nor do they have information on the samples numbers, so it is not clear how much difference there are between the lines.
With a sufficient number of data, it would be possible to discuss statistically significant differences and correlations between expression levels of BcCUC2 and the strength of the morphological phenotype. The current manuscript is weak on these points, which reduces the reliability of the data.
Minor points
- There are no information about accession numbers of gene sequences used in this study.
- Fig. 3C
There are differences of BcCUC2 expression also at non-leaf tissues between line 001 and 082. Are there any morphological differences at these tissues?
- Fig. 5B
What is mCherry? Is it NLS-mCherry as a nuclear marker?
- Fig. 6A
"CaMV 35S terminator" or just "35S ter" is more common description rather than "CaMV 35S poly(A)".
- Fig. 6D
"OE5" is lacking in the photo
- Fig. 6C
It is difficult to recognize the leaf morphological difference. Showing close-up view or Black and White images would be helpful to increase visibility of outlines.
-
Round 2
Reviewer 2 Report
>> Point 1: Little insight into regulatory mechanism of BcCUC2 (My original comment omitted)
> Response 1: Thanks for your comments and advice. The specific reason for the difference in leaf morphology and BcCUC2 expression profiles between the lines 001 and 082 are not clear. As Reviewer’s consideration, the expression differences between the lines can be influenced by the differences in promoter sequence of BcCUC2, miR164 level or auxin levels. Hence, we will analyze the promoter sequence of BcCUC2, miR164 level or auxin levels between the lines 001 and 082 in the future, to elucidate the molecular mechanism of leaf morphology and BcCUC2 expression difference between the lines.
I understood the authors' scope. Then I recommend the authors to include short discussion on what can makes the expression profile different between line "001" and "082".
>> Point 2: The link is unclear between BcCUC2 expression profiles and the leaf morphology. (My original comment omitted)
> Response 2: Thanks for your suggestions. Considering the influence of auxin level, we will analyze the auxin content of line 001 and 082 in the future, to investigate the concrete causes of differences in leaf margin morphology between lines. It is possible that the distribution and responsiveness of auxin or some potential genes play important roles in the regulation of leaf serration. The non-significant difference of BcCUC2 expression level between 001 and 082 was consistent with the slightly serrate development of overexpression lines to some extent. Hence, we speculated that BcCUC2 have a role in the early step of leaf serration. We are sorry for the imprecise discussion, the statements have been rephrased in lines 207-209.
I don't think the authors' explanation is sufficient. Why the overexpression in Arabidopsis resulted in slight enhancement of serration if BcCUC2 has strong functionality to highly enhance the serration with only 1.6 fold higher expression?
Pak-choi line "082" may have another molecular characteristics to enhance serration in addition to the BcCUC2 expression though I agree it is possible there is a small contribution from BcCUC2 to the highly serrated leaf morphology of line "082". Alternatively, BcCUC2 may be less functional in Arabidopsis than in Pak-choi.
I think there should be some revision in Discussion.
>> Point 3: Data sizes are not clearly described (My original comment omitted)
> Response 3: Thanks for your comments. We are very sorry for our negligence. Three biological repeats were used in this study, about 20 and 32 seedlings were used for different tissues and treatment (Figures 3 and 4), respectively, the related description have been added in lines 91-92. Three biological repeats were also used in subcellular localization (Fig. 5).
I appreciate the authors revision.
By the way, I have a concern on Figure3. It says "Statistical significance (ANOVA) is designated by * p<0.05, ** p<0.01.". I wonder if ANOVA really outputs each p value of tissue-by-tissue comparison. I think t-test is standard in the case of Fig.3C comparisons.
>> Point 4: Lack of quantification of morphological traits. (My original comment omitted)
> Response 4: Thanks for your comments. Lines 001 and 082 were obtained through multiple generations selfing, and the phenotype were basically consistent in the same genotype. There are 20 plants for each line were used for phenotype observation, and the phenotype were also consistent in each line. Considering the Reviewer’s suggestion, added the related description in lines 141-142. We are sorry for our negligence, the relevant indices for the serration level of leaves and the number of branches have not been quantified. We will make improvements in further related studies.
I appreciate the authors' revision of Fig.6C. However, the number of branches are not obvious in Fig.6D. It would be better to indicate branch numbers of the plant shown in the photo. Or, it would be okey to add arrowheads or any symbols to indicate each branch visible in the photo.
Minor points
>> Point 1: There are no information about accession numbers of gene sequences used in this study.
> Response 1: Thanks for your reminder. We have added the accession numbers in Table A2.
I could not find accession numbers in Table A2.
>> Point 2: Fig. 3C There are differences of BcCUC2 expression also at non-leaf tissues between line 001 and 082. Are there any morphological differences at these tissues?
> Response 2: Thanks for your question. The petiole length and leaf angle of line 082 was longer and bigger than line 001, respectively, with the exception of leaf serration.
Thank you for the information.
I asked this because it would be interesting to discuss the link between BcCUC2 expression and morphology. However, it is not necessary if the authors do not intend to discuss this point.
>> Point 3: Fig. 5B: What is mCherry? Is it NLS-mCherry as a nuclear marker?
> Response 3: Thanks for your comment. We are sorry for our negligence, mCherry is a nuclear marker, we have added the information in lines 238-240.
I appreciate the authors' revision.
>> Point 4: Fig. 6A: "CaMV 35S terminator" or just "35S ter" is more common description rather than "CaMV 35S poly(A)".
> Response 4: We are sorry for our negligence, "CaMV 35S poly(A)" should be "CaMV poly(A) signal", we have changed it in Fig. 6A.
A gene terminator and a poly(A) signal are different. Usually, a poly(A) signal is a short sequence like "AAUAAA". A terminator is a longer region including a poly(A) signal and other regulatory elements. According to the illustration in Fig.6A, I guess "CaMV 35S terminator", "35S ter" or other similar description is correct here.
>> Point 5: Fig. 6D: "OE5" is lacking in the photo.
> Response 5: Thanks for your comment. For the consideration of picture width and layout, OE5 was not placed here for photograph in the previous study.
I understood the situation. It would be good to include another photo of OE5 in appendix if authors have any photo though it is not very necessary.
>> Point 6: Fig. 6C: It is difficult to recognize the leaf morphological difference. Showing close-up view or Black and White images would be helpful to increase visibility of outlines.
> Response 6: Thanks for your suggestion, we have added the local enlarged drawing of leaves in Fig. 6C.
I appreciate the authors' revision.
another minor point
L308: I guess "001" instead of "082" is correct.
Though I am not an English native, I think English proofreading is needed. There are unnatural English expressions.
For example, the starting sentence "Abstract: Leaf margin serration is one of morphological characteristics, the CUC2 (CUP-SHAPED COTYLEDON 2) gene plays an important role in the outgrowth of leaf teeth and enhances leaf serration via suppression of growth in the sinus. "
In this sentence, it seems like the second half explains the first half of the sentence, but the sentence breaks are unclear. I think "characteristics." Instead of "characteristics," would be natural.
The revised part "The transcript levels of BcCUC2 were significantly higher in the young leaves of cultivar ‘082’ than cultivar ‘001’, while were decreased in mature leaves, speculating that BcCUC2 have a role in the early step of leaf serration." is unnatural. The subject who speculate should be the author but this says as if "The transcript speculates that BcCUC2 have a role". Maybe "possibly implying that" or "suggesting a possibility that" instead of "speculating" would be natural.
English quality is important for readers to grasp the exact meaning of the sentences and the strength of author's claim.
